# Implementing a Symmetric Lightweight Cryptosystem in Highly Constrained IoT Devices by Using a Chaotic S-Box

**Badr M. Alshammari** [1], **Ramzi Guesmi** [2,3,*], **Tawfik Guesmi** [1], **Haitham Alsaif** [1] and **Ahmed Alzamil** [1]

1   College of Engineering, University of Ha'il, Ha'il 81481, Saudi Arabia; bms.alshammari@uoh.edu.sa (B.M.A.); tawfik.guesmi@istmt.rnu.tn (T.G.); h.alsaif@uoh.edu.sa (H.A.); aa.alzamil@uoh.edu.sa (A.A.)
2   ISLAI Béja, University of Jendouba, Béja 9000, Tunisia
3   Laboratory of Electronics and Information Technology, National Engineering School of Sfax, Sfax University, Sfax 3038, Tunisia
*   Correspondence: ramzi.guesmi@gmail.com

**Abstract:** In the Internet of Things (IoT), a lot of constrained devices are interconnected. The data collected from those devices can be the target of cyberattacks. In this paper, a lightweight cryptosystem that can be efficiently implemented in highly constrained IOT devices is proposed. The algorithm is mainly based on Advanced Encryption Standard (AES) and a new chaotic S-box. Since its adoption by the IEEE 802.15.4 protocol, AES in embedded platforms have been increasingly used. The main cryptographic properties of the generated S-box have been validated. The randomness of the generated S-box has been confirmed by the NIST tests. Experimental results and security analysis demonstrated that the cryptosystem can, on the one hand, reach good encryption results and respects the limitation of the sensor's resources, on the other hand. So the proposed solution could be reliably applied in image encryption and secure communication between networked smart objects.

**Keywords:** Internet of Things; lightweight cryptography; chaos; Boolean function; S-box; Hilbert curve; AES; wireless sensor



## 1. Introduction

### 1.1. Research Background and Motivation

The data security is considered to be one of the most critical issues; it is an indispensable requirement especially for the operations and transactions that are based on data. As a matter of fact, data encryption is required before transmitting the data into the network. Developing new technologies for IoT without considering security will make the privacy of users' data vulnerable. The integration of the IoT with the security protocols is therefore a challenge. Recently, security aspects in the Internet of Things are getting more and more attention. Sensor nodes are characterized with their limited capacities and therefore implementing solutions based on the actually security protocols makes the subject challenging. In order to guarantee security in IOT, communications should be encrypted. Unfortunately, security is in general considered to be complex. Its cost is even more noticeable because of the limited resources of the sensor nodes [1]. Therefore, lightweight cryptography proved to be suitable for limited resources in IoT. As an emerging paradigm, the smart city avails and takes advantage of the existence of a variety of promising solutions, such as artificial intelligence (AI), Internet of Things (IoT), big data analysis, and real-time control. Many research works studied smart cities from several perspectives [2]. The main problem that was mentioned in these recent research studies was "security". In [3], authors presented the concept of citizen's privacy where they detailed a model which distinguishes five dimensions: identity privacy, query privacy, location privacy, footprint privacy, and owner privacy. The security problem becomes much more serious due to the vulnerabilities of IoT devices. In [4], Abomhara et al. showed that IoT has resulted in the emergence of new

types of security threats and it is evident that sensors and devices used in the IoT network could be targeted with malware and be vulnerable to several attacks.

*1.2. Related Work*

For the last two decades, lightweight cryptography has received significant attention and this has increased further in the last five years. Cryptographic algorithms must be used in the communication channels between the sensors so as to provide security. However, because of the very low energy available and the size of ROM and RAM, the cryptographic algorithms shall be as "small" as possible. Several lightweight algorithms were presented and they drew attention to their high security level [5]. These algorithms are still resistant against different attacks and are still effective. During the design of the cipher, PRESENT, Bogdanov et al. focused on security and hardware efficiency [6]. PRESENT, a leading algorithm for a lightweight block cipher, has a competitive requirement in compact hardware implementation compared with other algorithms. Several attempts have been suggested but none has succeeded in cryptanalysing the algorithm [7,8]. In [9], the blockcipher named CLEFIA was proposed. 128, 192 and 256 bits are the lengths key which are supported by CLEFIA. The authors explained in their paper that achieving a high level of efficiency in both hardware and software implementation as well as maintaining high levels of security was a challenge. In their paper (2008), Tsunoo et al. presented an impossible differential attack of CLEFIA [10]. Another lightweight algorithm, named LED, was described in 2011 [11]. The authors claimed that their solution can be adapted efficiently for lightweight hardware implementation. However an efficient attack has been found in [12]. Piccolo is the lightweight algorithm developed by Shibutani et al. [13]. It is a 64-bit blockcipher with keys of 80 and 128-bit. The authors showed that Piccolo ensures both a high level of security and an efficient implementation in hardware. They, also, made the point that the algorithm can resist against differential attacks and meet-in-the-middle attacks. An efficient attack has been found in [14]. In [15], a new cryptosystem named PRINCE was proposed taking into account the latency during the implementation in hardware. Compared with known solutions, PRINCE enables the encryption of data within one clock cycle with a very competitive chip area. Banik et al. presented an algorithm named Midori. The algorithm, which is characterized by its low energy consumption and compact hardware implementation, [16] is based on two block ciphers Midori128 and Midori64 with block sizes equal to 128 and 64 bits, respectively. Key weaknesses and efficient attacks have been found [17,18]. In order to ensure better results, several algorithms took advantage of the strength of AES algorithm. Sungha Kim and Ingrid Verbauwhede showed in their paper that implementing the Rijndael algorithm using 16 registers improves the efficiency over 40% in both speed and code size. They also showed that 128-bit size of input block is optimal for Rijndael implementation on 8-bit microcontroller [19]. In [20], authors tried to identify the candidates of blocks ciphers which are suitable for wireless sensor networks (WSNs), by constructing an evaluation framework. Because of the security properties, the storage and energy efficiency, authors selected the most appropriate ciphers for WSNs: Skipjack, MISTY1 and Rijndael. Andrea Vitaletti and Gianni Palombizio tried to answer the question: is speed the main issue? In their paper [21], they revealed that the developers of encryption algorithms for WSNs need to give priority to memory occupation and energy efficiency over speed. The designers showed that their AES-based scheme can be a solution for data encryption and for end-to-end encryption. They also developed a nesC module On the operating system TinyOS that enables users to encrypt messages at the application layer. An implementation of AES algorithm on MOTE-KIT 5040 was presented in [22]. The main contribution of the paper is the development of an encryption algorithm based on multi-space random key pre-distribution system for wireless sensor network. In [23], an implementation of an encryption algorithm like-AES was proposed. The authors aim to ensure sufficient levels of security so as to improve data privacy in WSN networks. As a basic nonlinear component of symmetric algorithms, substitution boxes (S-boxes) are the core component of the AES algorithm. In modern cryptography, image encryption algo-

rithms essentially make use of S-boxes to be able to strengthen substitution phases [24–28]. Recently, many powerful S-boxes have been generated on the basis of chaos functions because of their nonlinear property. A one-dimensional discrete-space chaotic system was proposed in [29]. The authors detailed in their paper the design algorithm of a new S-box using the proposed chaotic map which is based on the multiplication of integer numbers and circular shift. In [30], authors used a hybrid chaotic map in order to design an image encryption scheme. The proposed chaotic map displayed good cryptographic properties. In [31], authors designed a chaotic encryption system to generate a new S-box. In [32], authors presented a novel algorithm for designing a strong S-box. Their approach was based on cellular automata, and a fractional linear transformation over the Galois field.

### 1.3. Contributions of the Work

Most of the previous papers used simulations to assess the efficiency and strength of their algorithms. We think that a real implementation of encryption algorithms on real sensors provides more realistic results. Thus, our goal is to develop a lightweight scheme which is based on a modified AES algorithm and then implement it on a real wireless sensor. This modified algorithm is based on a new chaotic S-box that showed good cryptographic properties and a high level of randomness. In this paper, two main contributions are detailed. The first one is the drawing of a new strong S-box that passed all NIST tests. This S-box is essentially based on the generation of chaotic Boolean Functions aiming at reinforcing the nonlinear aspect. The Hilbert curve, which is a type of space-filling curves, was used to redistribute values in the S-box to ensure a high level of randomness. Our second contribution is the implementation of the encryption algorithm into a real sensor node characterized by limited capacities. In order to validate the strength of our encryption algorithm, a practical experiment was setup by encrypting a grayscale image on physical Wireless Sensor. This algorithm was implemented on Crossbow TelosB mote [33]. The use of the AES algorithm is justified by its inclusion in the IEEE 802.15.4 [34] standard. It is also the standard encryption protocol for ZigBee making it ideal for securing data exchange in wireless sensor networks.

## 2. Preliminaries

### 2.1. Internet of Things

IoT can be defined as a paradigm that takes into account the considerable presence of various things that can communicate with each other through wireless and wired connections [35]. The rapid development of the classical Internet into the IoT is empowering the exploration of countless domains of utilities that were previously unimaginable [36]. IOT networks, especially WSNs, are generally composed of constrained objects that are handled by a non-constrained object. A mutual authentication is necessary between a given device and the device manager if the former is aimed to join a WSN. Then, a symmetric secured pipe is created between the communicating entities in order to secure the exchanged data. IoT has been criticized for developing rapidly without taking into account the profound security challenges it entails and the necessary regulatory changes it require [37].

To deal with the issue of security of IoT, it is essential to first, understand all the building blocks of IoT. Then, one should identify each block's area of vulnerability and finally explore the necessary technologies to counter each weaknesses. Things, gateways, network infrastructure and cloud infrastructure are the main components of an IoT architecture [38]. The IOT architecture is illustrated in Figure 1.

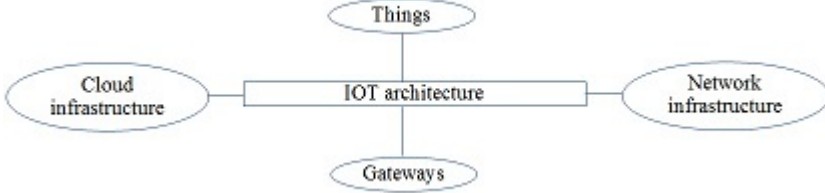

**Figure 1.** IOT Architecture.

## 2.2. Boolean Functions

One of the most interesting methods of drawing symmetric key algorithms is Boolean functions. Their properties play a key role in cryptography where an S-box (substitution-box) is a basic component of symmetric key algorithms which performs substitution. A Boolean function on n variables takes the form $\mathbb{F}_2^n$ into $\mathbb{F}_2$. S-boxes can therefore be defined as (n, m) Boolean functions [39]. The generated S-box should have good cryptographic properties such as nonlinearity, bijection, the strict avalanche criterion, the output bits independence criterion (BIC) and the equiprobable input/output XOR distribution. A detailed description can be found in [40].

## 2.3. NIST Statistical Test Suite

To evaluate the different aspects of the randomness of binary sequences, cryptographers refer to the 15 NIST tests [41]. These tests evaluate different types of non-randoms that could exist in a sequence.

In this paper, NIST Statistical Test Suite will be called to study the randomness of the generated S-boxes, the chaotic PRNG and the ciphered images.

## 2.4. The Lorenz System

The high sensitivity to initial values is an important feature of chaotic systems. The Lyapunov exponent provides a quantitative description of the initial state sensitivity of a chaotic system [42]. The chaotic behavior of the Lorenz map is described by the following equation [43–45]:

$$\begin{cases} \dot{x} = a(y - x) \\ \dot{y} = cx - y - xz \\ \dot{z} = xy - bz \end{cases} \tag{1}$$

where: the system state $(x,y,z)$, the system parameters $(a,b,c)$. The Lorenz attractor and the Lyapunov exponents are illustrated in Figures 2 and 3 respectively.

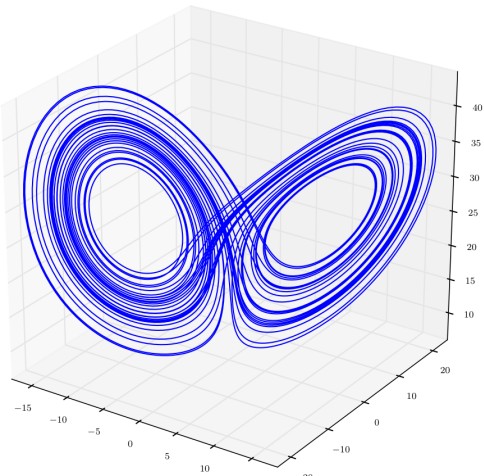

**Figure 2.** The Lorenz attractor.

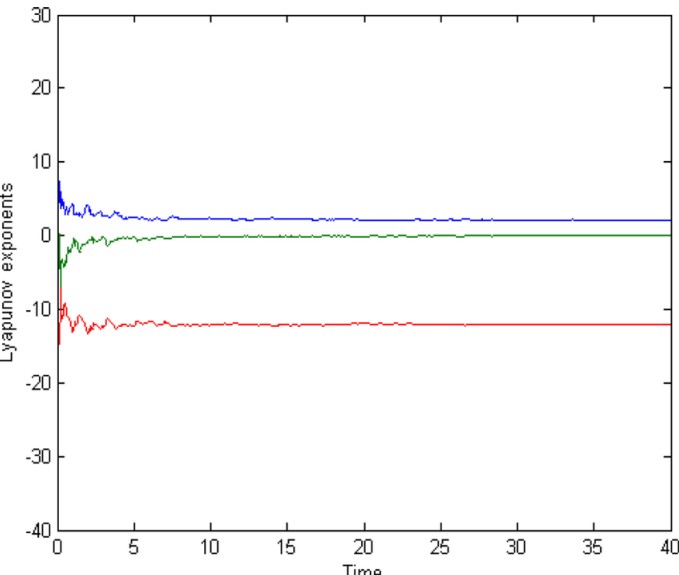

**Figure 3.** The lyapunov exponents.

The main usefulness of the Lorenz system is to generate chaotic binary sequences in order to create the Boolean Functions.

*2.5. SHA-2*

The SHA-2 function was used in order to generate 256-bit external secret key K [46,47]. This secret key K will strengthen the proposed cryptosystem by increasing the complexity of the encryption algorithm to $2^{256}$. Therefore it can withstand the brute-force attack. K is split into 8-bit blocks as follows.

$$K = k_1, k_2, k_3, ..., k_{32}. \tag{2}$$

The initial values can be derived as follows.

$$x_0 = x_0' + (k_1 \oplus k_2 \oplus k_3 \oplus ... \oplus k_{11})/256 \tag{3}$$

$$y_0 = y_0' + (k_{12} \oplus k_{13} \oplus k_{14} \oplus ... \oplus k_{22})/256 \tag{4}$$

$$z_0 = z_0' + (k_{23} \oplus k_{24} \oplus k_{25} \oplus ... \oplus k_{32})/256 \tag{5}$$

where $x_0'$, $y_0'$ and $z_0'$ are the initial given values.

**3. Scan Methodology**

In [48,49], Giuseppe Peano and David Hilbert demonstrated that Space-filling curves are fractal objects. They formulate curves that visit every point in a unit square. The Scan represents a family of two-dimensional spatial accessing methodology to generate a large number of scanning paths [50]. The Hilbert curve $H_{2^n}$, for $n \geq 1$, is a fractal structure that is generated by the following recursive production rule [51]:

$H_{2^n}$ $\rightarrow \text{rightRot}(H_{2^{n-1}}) \rightarrow U \rightarrow H_{2^{n-1}} \rightarrow R \rightarrow H_{2^{n-1}} \ D \rightarrow \text{leftRot}(H_{2^{n-1}}) \rightarrow$

**rightRot($H_2$)** $\rightarrow U \rightarrow R \rightarrow D \rightarrow$ -

**rightRot($H_{2^n}$)** $\rightarrow R \rightarrow U \rightarrow L \rightarrow$

**leftRot($H_2$)** $\rightarrow L \rightarrow D \rightarrow R \rightarrow$

where D, U, L and R indicate the directions taken by the curve (Down, Up, Left and Right).

## 4. Hilbert Curve Scan Pattern

The scan path of Hilbert curve can be drawn either from right bottom (RB), left bottom (LB), right top (RT) or left top (LT) of the square grid [52,53]. The proposed construction method of the S-box is based on this scan path. The application of Hilbert curve scan pattern in this work is to redistribute values in the initial generated S-box. The representation of Hilbert curve using matrices is performed recursively via successive approximations which are called an order for that curve. The Orders ($n$) one through three of the Hilbert curve are shown in Figure 4.

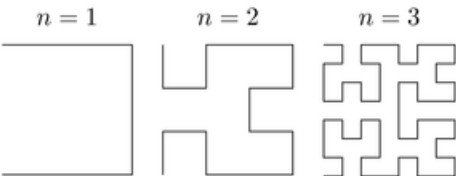

**Figure 4.** Hilbert Curve.

Figure 5 shows that values from 0 to 3 are provided by the first order Hilbert curve and values from 0 to 15 are provided by the second order Hilbert curve.

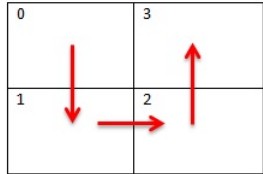 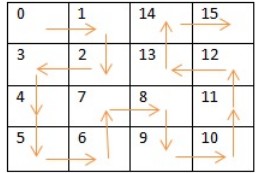

**Figure 5.** Hilbert Curve values.

The idea of using this Hilbert curve comes from the fact that we can develop an algorithm that describes it easily. This algorithm can be very interesting to redistribute values in a matrix. For example, suppose that the binary representation for a random cell N on the curve of order $o$: $H_o(N) = 111001010...11$. The four children $n_0$, $n_1$, $n_2$ and $n_3$ of $H_o(N)$ of order $o + 1$ will be represented as follows:

$$
\begin{aligned}
H_{o+1}(n_0) &= CONCAT(H_o(N), 00) \\
H_{o+1}(n_1) &= CONCAT(H_o(N), 01) \\
H_{o+1}(n_2) &= CONCAT(H_o(N), 10) \\
H_{o+1}(n_3) &= CONCAT(H_o(N), 11)
\end{aligned}
\tag{6}
$$

## 5. Modified AES S-Box Generation

An S-Box takes $m$ input bits and transforms them into $m$ output bits where $n$ can be different to $m$. It is called ($m \times n$) S-box and is implemented as a LookUp Table (LUT). The basic function of an S-Box is transforming one byte of input data into another one. It is specifically designed to be resistant to linear and differential cryptanalysis.

### 5.1. The Main Idea

The generating algorithm of the S-box is based on creating a set $S$ of 2048 bits by using a chaotic map. Then, 8 subsets of Boolean functions are obtained from the main set $S$. The flowchart of the construction method is shown in Figure 6.

### 5.2. Generate the Chaotic Boolean Functions

1.　Generate a chaotic set $S$ of 2048 bits.

- Define an empty binary set $S$.

- Iterate Equation (1) for 200 times to get rid of the transient effect by using $x_0, y_0, z_0$.
- Iterate Equation (1) for 2048 times and denote the current state value as $x', y', z'$. $x$ is obtained by Equation (7) and inserted in $S$.

$$x = mod(floor(10^4 * x'), 2) \tag{7}$$

2. Get $m$ Boolean functions by dividing the set $S$ into 8 binary subsets of 256 bits.
3. Create the initial S-box.
4. Adjustment: Each element in the S-box must be unique. To guarantee this property, an adjustment of the S-box is needed.

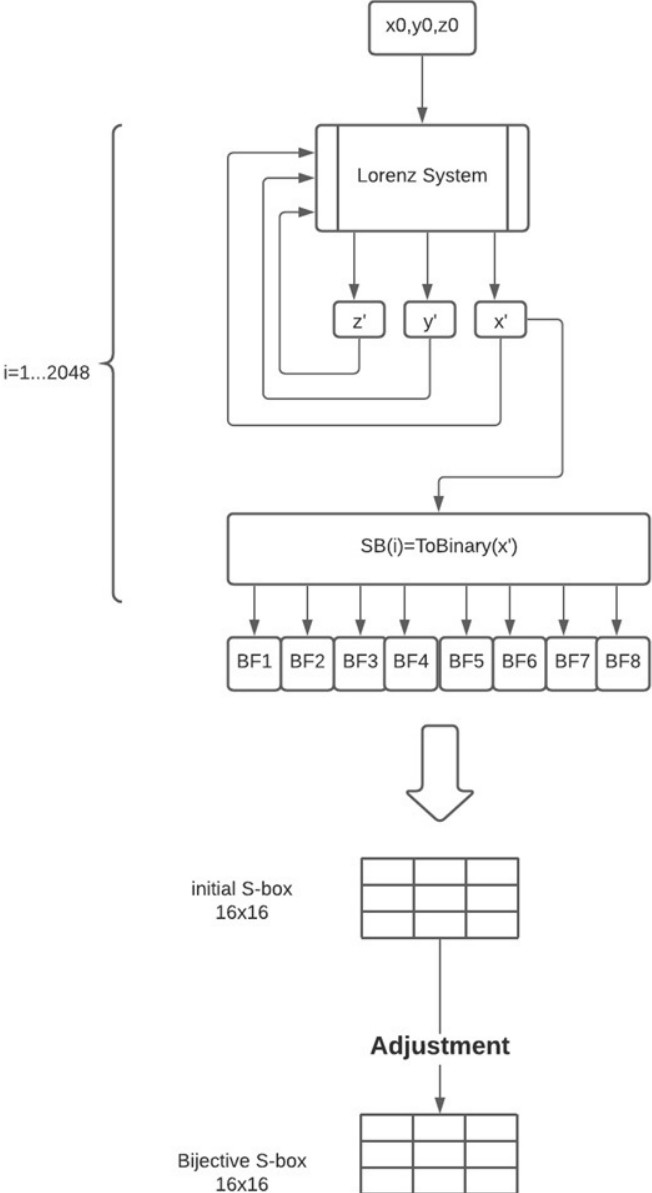

**Figure 6.** Flowchart of the initial S-box.

### 5.3. Generate a Cryptographically Strong S-box

In this part, the initial S-box which was obtained above will be improved in order to produce a better S-box. Improving the algorithm is based on the Hilbert curve that will be used to permute values of the initial S-box. This permutation step will be iterated 1000 times. At each iteration we calculate the nonlinearity value of the generated S-box. Whenever we

find a high value of nonlinearity, we save the corresponding S-box. The improvement of the algorithm of the S-box is described in Figure 7.

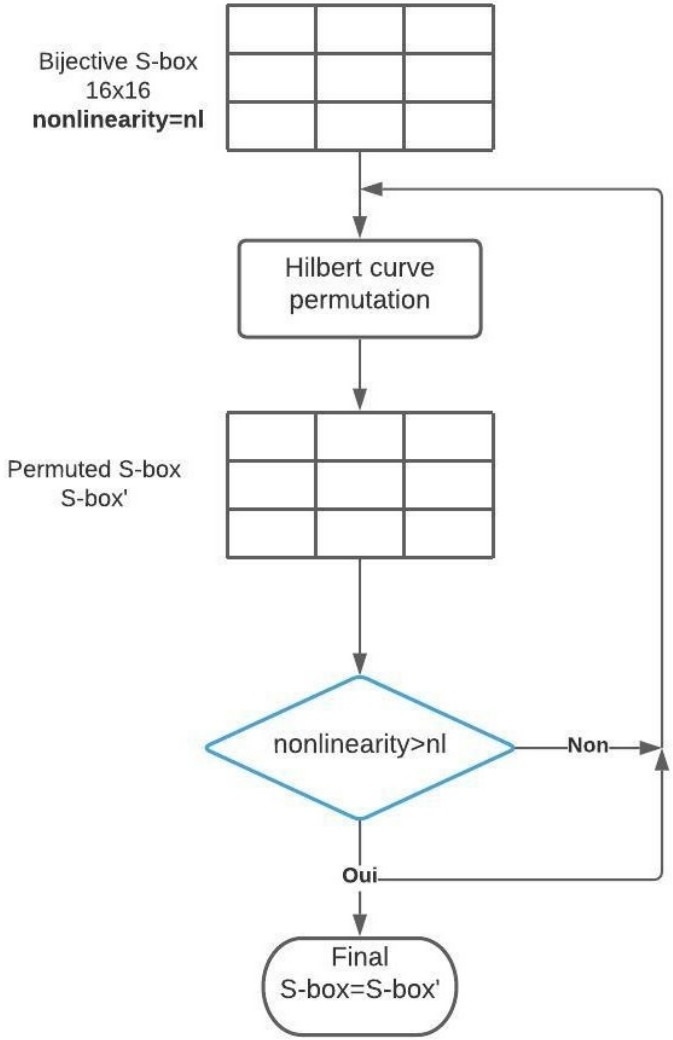

**Figure 7.** Flowchart of the improvement algorithm.

### 5.4. Experimental Results and Cryptographic Properties of the Chaotic S-box

The initial values of the Lorenz system are $x_0 = 0, y_0 = 1, z_0 = 1.05$. The initial generated S-box is illustrated in Table 1. We keep permuting the values of the S-box and we calculate the nonlinearity of the generated S-boxes at each iteration until we get the S-box with the highest value of nonlinearity. Table 2 shows the best S-box with a value of nonlinearity equal to 107. Table 3 shows the inverse S-box which is simply the S-box run in inverse. It is used for the decryption algorithm. After executing our algorithm, nonlinearity value increased from 104 (iteration number 0) to 107 (iteration number 602). The evolution of the nonlinearity value is illustrated in Figure 8. The improvement of the value of nonlinearity is achieved thanks to the permutation step ensured by the Hilbert curve-based scan pattern.

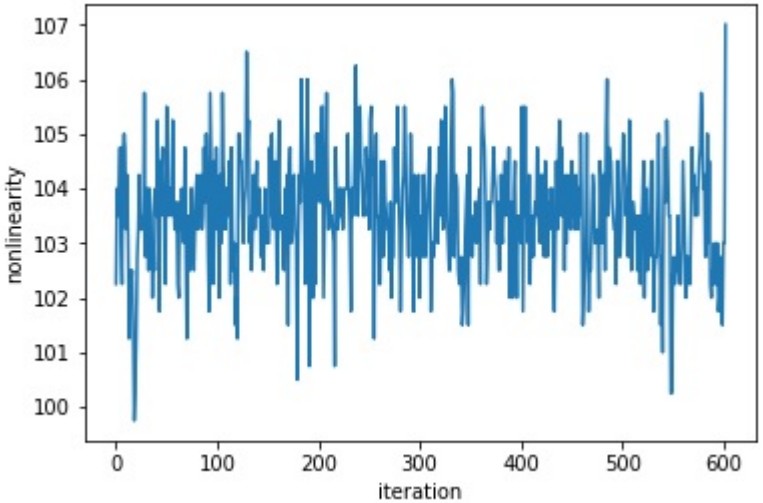

**Figure 8.** The nonlinearity evolution.

**Table 1.** The generated initial S-Box.

| | | | | | | | | | | | | | | | |
|---|---|---|---|---|---|---|---|---|---|---|---|---|---|---|---|
| 49 | 170 | 30 | 184 | 28 | 232 | 128 | 51 | 80 | 99 | 130 | 154 | 0 | 191 | 94 | 233 |
| 223 | 106 | 255 | 120 | 7 | 162 | 254 | 68 | 81 | 16 | 131 | 72 | 123 | 192 | 204 | 234 |
| 27 | 66 | 227 | 10 | 216 | 41 | 196 | 89 | 116 | 102 | 133 | 75 | 164 | 197 | 206 | 235 |
| 18 | 5 | 190 | 12 | 19 | 150 | 43 | 45 | 60 | 246 | 173 | 119 | 113 | 112 | 225 | 238 |
| 211 | 193 | 24 | 253 | 181 | 46 | 229 | 146 | 127 | 108 | 134 | 23 | 176 | 198 | 74 | 239 |
| 185 | 100 | 8 | 105 | 212 | 17 | 58 | 15 | 243 | 111 | 97 | 156 | 250 | 110 | 207 | 33 |
| 35 | 132 | 169 | 1 | 63 | 83 | 136 | 54 | 165 | 114 | 138 | 85 | 167 | 199 | 209 | 25 |
| 70 | 221 | 21 | 44 | 59 | 163 | 142 | 36 | 82 | 115 | 139 | 157 | 178 | 200 | 168 | 240 |
| 2 | 61 | 26 | 140 | 194 | 47 | 137 | 152 | 84 | 64 | 143 | 158 | 174 | 62 | 215 | 241 |
| 188 | 87 | 96 | 210 | 29 | 208 | 69 | 76 | 86 | 141 | 145 | 236 | 175 | 201 | 14 | 242 |
| 159 | 101 | 166 | 155 | 109 | 48 | 144 | 73 | 90 | 248 | 147 | 220 | 179 | 226 | 171 | 245 |
| 224 | 22 | 92 | 13 | 237 | 53 | 42 | 122 | 91 | 117 | 149 | 31 | 182 | 203 | 218 | 129 |
| 20 | 6 | 93 | 230 | 189 | 57 | 65 | 180 | 98 | 247 | 153 | 148 | 214 | 9 | 219 | 249 |
| 4 | 55 | 11 | 222 | 37 | 244 | 205 | 77 | 104 | 121 | 88 | 217 | 183 | 202 | 3 | 251 |
| 39 | 52 | 50 | 118 | 34 | 213 | 32 | 78 | 151 | 125 | 135 | 160 | 186 | 95 | 228 | 252 |
| 195 | 172 | 71 | 103 | 38 | 126 | 67 | 79 | 177 | 124 | 107 | 161 | 187 | 40 | 231 | 56 |

**Table 2.** The S-Box.

| | | | | | | | | | | | | | | | |
|---|---|---|---|---|---|---|---|---|---|---|---|---|---|---|---|
| 31 | DF | 77 | 67 | 9D | D8 | B6 | 11 | 94 | 32 | E2 | E0 | 16 | AB | 6D | F5 |
| 33 | A0 | 02 | 01 | 7C | E3 | 84 | DD | FE | 69 | 71 | 44 | 52 | 6A | 5A | 3E |
| 58 | E9 | 97 | BE | 1F | C1 | FA | 13 | 8C | A7 | 34 | A1 | 48 | 7D | F4 | 4B |
| 10 | 39 | E5 | C2 | D0 | F8 | 8B | 43 | D6 | F0 | AA | B9 | 4E | 57 | E6 | 1C |
| D3 | 0B | 03 | 81 | 5F | D4 | 68 | 23 | 5E | 61 | FD | CD | DA | 29 | 9F | EA |
| B8 | 41 | B2 | EB | 14 | 8F | 51 | 26 | 8A | 27 | 00 | C3 | 5B | 5D | 5C | 3D |
| 73 | 15 | 4A | 85 | C5 | B3 | A2 | 22 | 8E | 6C | 17 | A4 | C6 | A8 | 86 | A3 |
| 50 | 80 | 70 | AF | 87 | 6B | B1 | 18 | 07 | 30 | 62 | 08 | 6E | 1A | 45 | 55 |
| 1E | 47 | 35 | 3C | EC | 3B | 98 | 99 | B5 | 56 | F7 | 66 | 2F | 1D | 72 | 64 |
| 3A | 65 | 78 | 91 | EE | 63 | 88 | 19 | 92 | E1 | 28 | 6F | A6 | FF | A5 | 7A |
| 0C | C0 | 0D | F9 | 9E | 3F | 49 | B0 | F2 | AD | 74 | 95 | 83 | 93 | 89 | 0A |
| 04 | BC | C7 | 96 | BB | BA | 25 | 54 | 46 | CE | C9 | CB | 4D | B4 | BF | 42 |
| 2D | C8 | 36 | A9 | 0E | 53 | DB | CC | EF | 2C | 75 | F3 | 2B | 2A | D1 | C4 |
| 37 | D5 | 0F | DC | CA | D2 | 60 | 4C | FC | 59 | 2E | 24 | AC | E8 | 38 | B7 |
| 05 | 9A | FB | 12 | 90 | 82 | CF | 20 | 06 | DE | 9C | F1 | 79 | 7E | 76 | E7 |
| E4 | 40 | 1B | 4F | BD | 7B | D7 | D9 | AE | 21 | 9B | ED | F6 | 7F | 09 | 8D |

**Table 3.** The inverse S-Box.

| | | | | | | | | | | | | | | | |
|---|---|---|---|---|---|---|---|---|---|---|---|---|---|---|---|
| 49 | 51 | 88 | 16 | 211 | 184 | 115 | 80 | 30 | 58 | 12 | 4 | 45 | 55 | 5 | 228 |
| 223 | 160 | 233 | 57 | 11 | 65 | 21 | 128 | 71 | 101 | 192 | 188 | 200 | 213 | 154 | 64 |
| 119 | 2 | 151 | 229 | 3 | 178 | 74 | 112 | 53 | 120 | 13 | 199 | 54 | 15 | 251 | 27 |
| 103 | 1 | 190 | 194 | 129 | 235 | 133 | 175 | 60 | 145 | 249 | 150 | 169 | 220 | 18 | 79 |
| 157 | 124 | 31 | 208 | 95 | 20 | 197 | 135 | 236 | 238 | 158 | 187 | 14 | 202 | 144 | 189 |
| 216 | 227 | 193 | 248 | 212 | 143 | 179 | 107 | 59 | 99 | 63 | 186 | 83 | 210 | 130 | 123 |
| 182 | 132 | 250 | 139 | 104 | 81 | 162 | 177 | 152 | 136 | 73 | 37 | 219 | 96 | 207 | 215 |
| 17 | 221 | 19 | 67 | 35 | 38 | 34 | 24 | 153 | 25 | 176 | 84 | 204 | 76 | 32 | 217 |
| 148 | 254 | 140 | 214 | 94 | 138 | 142 | 7 | 181 | 146 | 242 | 70 | 239 | 252 | 6 | 174 |
| 50 | 105 | 167 | 240 | 97 | 39 | 108 | 48 | 86 | 225 | 173 | 206 | 44 | 89 | 222 | 33 |
| 226 | 113 | 52 | 170 | 253 | 0 | 23 | 98 | 247 | 40 | 116 | 201 | 117 | 46 | 156 | 155 |
| 224 | 68 | 161 | 185 | 205 | 195 | 164 | 8 | 102 | 111 | 149 | 203 | 243 | 36 | 241 | 237 |
| 22 | 82 | 72 | 78 | 218 | 91 | 198 | 110 | 47 | 166 | 131 | 77 | 43 | 172 | 121 | 246 |
| 171 | 106 | 125 | 87 | 41 | 93 | 168 | 26 | 29 | 255 | 147 | 180 | 42 | 232 | 126 | 127 |
| 109 | 90 | 244 | 230 | 159 | 92 | 134 | 69 | 114 | 165 | 137 | 191 | 209 | 56 | 118 | 9 |
| 245 | 62 | 75 | 28 | 234 | 61 | 163 | 85 | 100 | 122 | 10 | 66 | 196 | 183 | 231 | 141 |

The cryptographic properties of the S-box are presented as follows:

### 5.4.1. Bijectivity

The output values of the generated S-box are in the interval [0, 255]. Therefore, the S-box satisfies the requirement of bijectivity.

### 5.4.2. Nonlinearity

The average value of nonlinearity is equal to 107 Table 4. A comparison of the nonlinearity value of our s-box with others is illustrated in Table 5.

**Table 4.** Nonlinearity and SAC.

| S-Box | 1 | 2 | 3 | 4 | 5 | 6 | 7 | 8 | Avg. Nonlinearity | Avg. SAC |
|---|---|---|---|---|---|---|---|---|---|---|
| | 108 | 108 | 106 | 108 | 106 | 106 | 108 | 106 | 107 | 0.4932 |

**Table 5.** Comparison of the nonlinearity of chaotic S-box with others.

| S-Box | 1 | 2 | 3 | 4 | 5 | 6 | 7 | 8 | Nonlinearity |
|---|---|---|---|---|---|---|---|---|---|
| AES S-box | 112 | 112 | 112 | 112 | 112 | 112 | 112 | 112 | 112 |
| APA S-box | 112 | 112 | 112 | 112 | 112 | 112 | 112 | 112 | 112 |
| Gray S-box | 112 | 112 | 112 | 112 | 112 | 112 | 112 | 112 | 112 |
| [54] | 110 | 108 | 108 | 110 | 108 | 108 | 110 | 112 | 109.25 |
| [28] | 108 | 108 | 108 | 108 | 108 | 108 | 108 | 108 | 108 |
| [55] | 108 | 110 | 110 | 108 | 106 | 106 | 106 | 110 | 108 |
| [56] | 106 | 108 | 110 | 106 | 110 | 106 | 106 | 108 | 107.5 |
| **Our S-box** | **108** | **108** | **106** | **108** | **106** | **106** | **108** | **106** | **107** |
| [57] | 108 | 106 | 106 | 106 | 106 | 106 | 108 | 108 | 106.5 |
| [29] | 108 | 108 | 106 | 106 | 106 | 106 | 106 | 106 | 106.5 |
| [58] | 106 | 108 | 108 | 106 | 106 | 106 | 106 | 106 | 106.5 |
| [59] | 108 | 106 | 104 | 108 | 106 | 110 | 104 | 104 | 106 |
| Skipjack | 104 | 104 | 108 | 108 | 108 | 104 | 104 | 106 | 105.75 |
| Xyi S-box | 106 | 104 | 104 | 106 | 104 | 106 | 104 | 106 | 105 |
| [60] | 103 | 109 | 104 | 105 | 105 | 106 | 104 | 103 | 104.88 |
| [61] | 107 | 103 | 100 | 102 | 96 | 108 | 104 | 108 | 103.5 |
| [62] | 104 | 100 | 106 | 102 | 104 | 102 | 104 | 104 | 103.25 |
| [63] | 104 | 106 | 104 | 108 | 98 | 100 | 100 | 106 | 103.25 |
| Residue Prime | 94 | 100 | 104 | 104 | 102 | 100 | 98 | 94 | 99.5 |

### 5.4.3. Strict Avalanche Criterion: SAC

The dependence matrix is illustrated in Table 6 and its mean value is 0.4932 which is very close to the ideal value 0.5. A comparison with other dependence values obtained from other S-boxes is illustrated in Table 7.

**Table 6.** The dependence matrix.

| | | | | | | | |
|---|---|---|---|---|---|---|---|
| 0.5000 | 0.4531 | 0.4375 | 0.50000 | 0.4609 | 0.5000 | 0.5234 | 0.5078 |
| 0.5000 | 0.5000 | 0.5000 | 0.50000 | 0.5000 | 0.5000 | 0.5000 | 0.5000 |
| 0.5000 | 0.5000 | 0.5000 | 0.50000 | 0.4609 | 0.4687 | 0.4765 | 0.5000 |
| 0.4375 | 0.5000 | 0.4375 | 0.50000 | 0.5000 | 0.5000 | 0.4765 | 0.5000 |
| 0.5000 | 0.5000 | 0.5625 | 0.50000 | 0.4609 | 0.5000 | 0.4765 | 0.4921 |
| 0.4375 | 0.5468 | 0.5625 | 0.50000 | 0.5000 | 0.4687 | 0.5234 | 0.4921 |
| 0.5625 | 0.5000 | 0.5000 | 0.50000 | 0.4609 | 0.5000 | 0.5000 | 0.5078 |
| 0.4375 | 0.4531 | 0.5000 | 0.50000 | 0.5000 | 0.5000 | 0.4765 | 0.5000 |

**Table 7.** Comparison of the values of the dependence matrix of chaotic S-box with others.

| S-Box | Minimum Value | Maximum Value | Average Value |
|---|---|---|---|
| AES S-box | 0.480 | 0.528 | 0.504 |
| APA S-box | 0.472 | 0.526 | 0.499 |
| Gray S-box | 0.478 | 0.526 | 0.502 |
| **Our S-box** | **0.4375** | **0.5625** | **0.4932** |
| [28] | 0.406 | 0.578 | 0.492 |
| Skipjack | 0.464 | 0.534 | 0.499 |
| Xyi S-box | 0.470 | 0.536 | 0.503 |
| [60] | 0.398 | 0.570 | 0.506 |
| [61] | 0.390 | 0.585 | - |
| [62] | 0.421 | 0.593 | - |
| [63] | 0.375 | 0.593 | - |
| Residue Prime S-box | 0.470 | - | 0.502 |

### 5.4.4. Bits Independence Criterion: BIC

The mean value of BIC-nonlinearity is 102.2857 which means that the S-box validates the BIC property (Table 8).

**Table 8.** BIC -Nonlinearity criterion.

| | | | | | | | |
|---|---|---|---|---|---|---|---|
| - | 108 | 104 | 104 | 100 | 100 | 102 | 100 |
| 108 | - | 98 | 104 | 106 | 98 | 104 | 104 |
| 104 | 98 | - | 106 | 92 | 102 | 102 | 100 |
| 104 | 104 | 106 | - | 104 | 104 | 100 | 106 |
| 100 | 106 | 92 | 104 | - | 104 | 104 | 102 |
| 100 | 98 | 102 | 104 | 104 | - | 98 | 108 |
| 102 | 104 | 102 | 100 | 104 | 98 | - | 100 |
| 100 | 104 | 100 | 106 | 102 | 108 | 100 | - |

### 5.4.5. Nist Statistical Test Suite

To evaluate the randomness of the S-box, we applied all NIST tests. Table 9 demonstrates that the generated S-box has passed all tests.

**Table 9.** Randomness test of the S-box.

| Statistic Tests | $p$-Value for Cipher-Images |
|---|---|
| Frequency (Monobits) test | 0.99258 |
| Frequency test within a block | 0.543019 |
| Runs test | 0.859684 |
| Test for the longest run of ones in a block | 0.99889 |
| Binary matrix rank test | 0.481248 |
| Discrete Fourier transform (Spectral) test | 0.167931 |
| Non-overlapping template matching test | Success |
| Overlapping template matching test | 0.886589 |
| Maurer's "universal statistical" test | 0.895623 |
| Linear complexity test | 0.919679 |
| Serial test (1) | 0.773031 |
| Serial test (2) | 0.079182 |
| Approximate entropy test | 0.0920 |
| Cumulative sums | 0.675322 |
| Random excursions test | Success |
| Random excursions variant test | Success |
| Results | Pass |

## 6. The Proposed Image Encryption Scheme

Many factors and constraints on sensor nodes must be taken into consideration for the implementation of lightweight cryptography. We listed essentially: energy, memory, computational speed and communications bandwidth. Because of the execution time, power consumption depends on the processing speed. Hence, the number of computations that determines the processing speed becomes the index of lightness.

With regard to security, lightweight encryption is the adopted method of the overall system security. This work is based on the fact that the proposed cryptosystem needs to be based on an algorithm that shows a sufficient security level in modern cryptography. In the context of lightweight algorithms, there have been many proposed optimised implementations that improve the algorithms performance [64].

The implementation of an encryption scheme in a wireless sensor depends on two determining factors: the memory size and the energy consumption. A wireless sensor with a RAM size of 8 KB and a ROM size of 116 KB is not able to do the following actions simultaneously:

- store the TinyOS operating system,
- store the encryption algorithm,
- store the grayscale image to be encrypted and
- run the algorithm in order to generate the encrypted image.

Therefore, the proposed solution consists of developing a lightweight algorithm and implementing it in an XM1000 wireless sensor. In the following part, we will show that the encryption algorithm is able to encrypt grayscale images of large sizes. The flowchart of the main algorithm is illustrated in Figure 9.

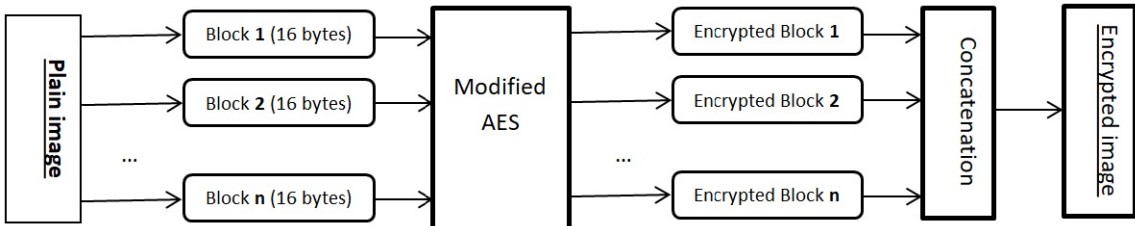

**Figure 9.** The flowchart of the main algorithm.

To implement the algorithm, we propose to encrypt the image by blocks of 16 bytes. We note that after running the code with larger blocks, the sensor gave us unstandard results and that is why we chose the 16 bytes. To encrypt the grayscale image, the main code is triggered via the Boot.booted() event. To get the ciphered image from the sensor, we divided it into blocks of $4 \times 16$ bytes and we sent each block apart. In our implementation, we used images of $50 \times 64$ bytes and a period of $5 \times 10^{-2}$ s. In TinyOs, Timer.startPeriodic() is the command to set the period and Timer.fired() is the event to send ciphered blocks. Finally, all received blocks are concatenated into one ciphered image. The generated S-box, which is used in the modified AES, is illustrated in Table 2.

## 7. Experimental Results

### 7.1. The Setup

As mentioned earlier, we chose to implement the solution on the XM1000 sensor which consists of the MSP430 microcontroller and the CC2420 radio chip. The code was implemented under the TinyOs operating system. TinyOS is an embedded operating system written in the nesC [65].

### 7.2. Security Analysis and Experimental Results

#### 7.2.1. Memory Consumption and Execution Times

Our algorithm was implemented under a physical sensor (XM1000 sensor), not a simulation, wich gave us real results. $50 \times 64$ is the size of the image we ciphered and the execution time of the algorithm is 230,399 milliseconds. The ROM consumption is 13,624 KB and the RAM consumption is 7826 KB.

#### 7.2.2. Information Entropy

Information entropy is an important feature of randomness and it is an indicator of the pixel values distribution. The equation to calculate it is illustrated in Equation (12):

$$H(m) = \sum_{i=0}^{2^n-1} p(m_i) log_2 \frac{1}{p(m_i)}. \tag{8}$$

$m$: information source, $p(m)$: probability of $m$ where $p(m)$ represents the probability of symbol $m$.

Table 10 shows the information entropy of the encrypted images.

**Table 10.** Entropy values.

| Images | Plain Image | Ciphered Image |
|---|---|---|
| Lena | 7.8055 | 7.9401 |
| Baboon | 7.6015 | 7.9290 |
| Boat | 7.2189 | 7.9433 |
| Clock | 7.0087 | 7.9366 |
| Peppers | 7.7645 | 7.9351 |
| Jelly-Beans | 7.1872 | 7.9331 |
| Splash | 7.5045 | 7.9460 |

The information entropy of the encrypted images is better than the information entropy of the encrypted ones. Therefore, the efficiency and security of the proposed cryptosystem are validated. We notice that all values of entropy are very close to 8 (Table 10). Therefore the probability of accidental information disclosure is minor.

#### 7.2.3. The Histogram Analysis

The histogram of an image measures the distribution of gray levels in the image [66,67]. Therefore, histograms have been plotted in order to evaluate the uniformity of the encrypted images [68]. Table 11 shows that the histograms of the encrypted images are uniform unlike

those of the plain images. Therefore the attacker cannot extract information from the encrypted image because the encryption algorithm damaged the original images' features.

**Table 11.** Histograms of the plain/encrypted images.

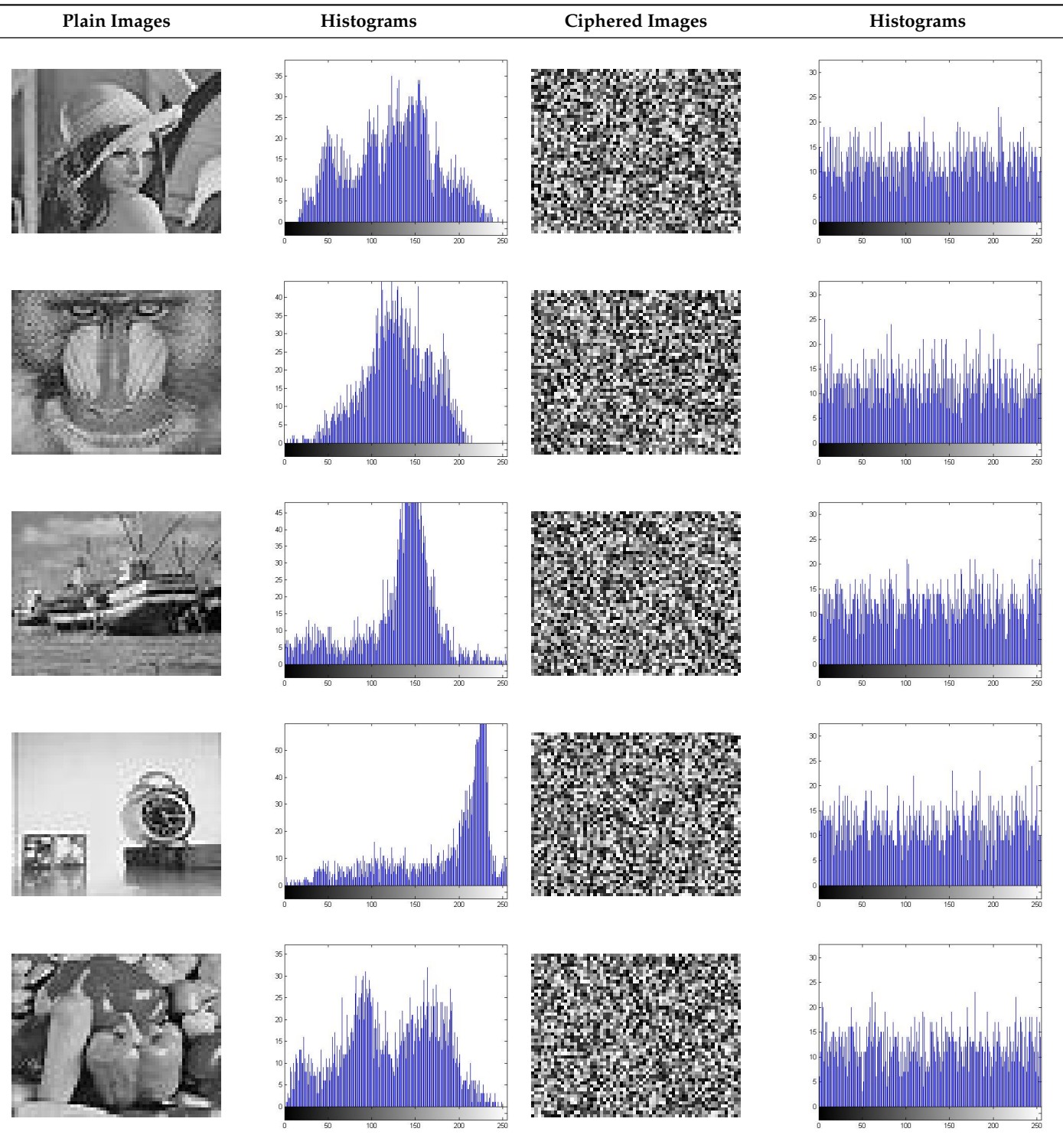

**Table 11.** *Cont.*

| Plain Images | Histograms | Ciphered Images | Histograms |
|---|---|---|---|

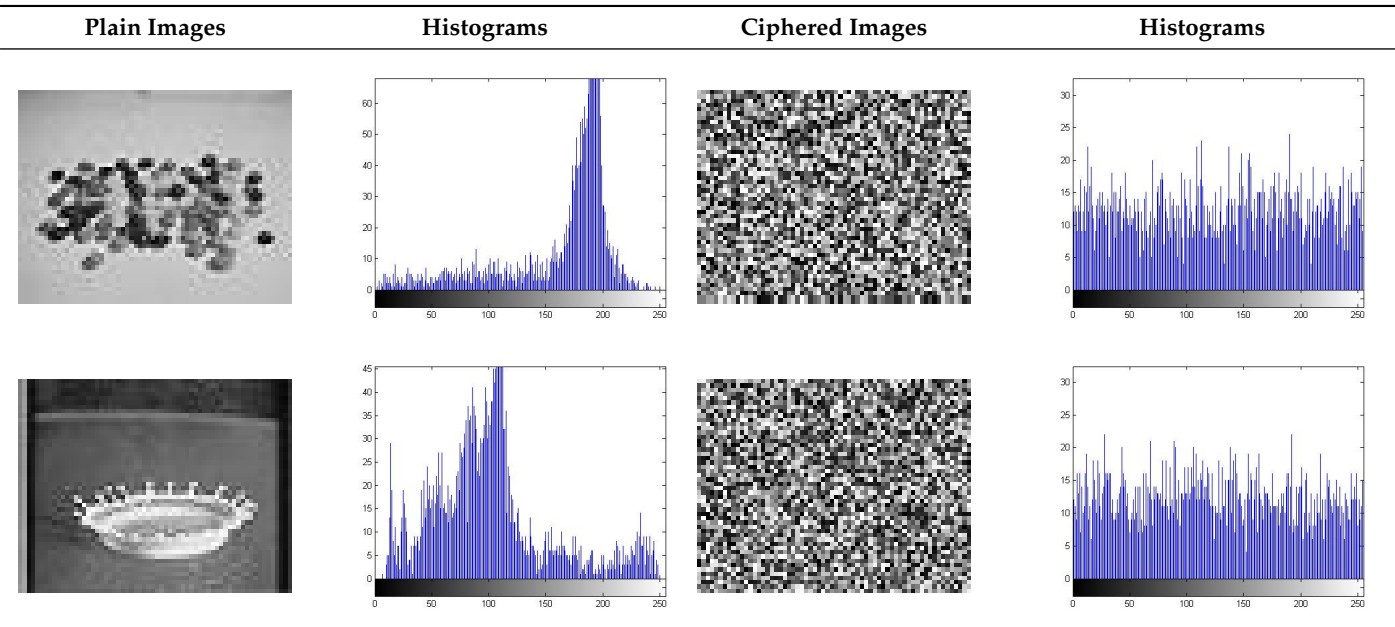

### 7.2.4. Correlation Coefficient Analysis

To measure the correlation between two adjacent pixels, horizontally, vertically and diagonally, developers of image encryption algorithms analysed correlation coefficients. The correlation coefficients of two adjacent pixels are calculated according to the following formula [69,70]:

$$r_{xy} = \frac{cov(x,y)}{\sqrt{D(x)}\sqrt{D(y)}}, \qquad (9)$$

where

$$cov(x,y) = \frac{1}{N}\sum_{i=1}^{N}(x_i - E(x))(y_i - E(y)), \qquad (10)$$

$$E(x) = \frac{1}{N}\sum_{i=1}^{N}x_i, \qquad (11)$$

$$D(x) = \frac{1}{N}\sum_{i=1}^{N}(x_i - E(x))^2. \qquad (12)$$

where $x$ and $y$ are gray level values of two adjacent pixels. $N$ is the total number of the selected pixels, $E(x)$ is the mean values of $x_i$ and $E(y)$ is the mean values of $y_i$. Correlation coefficients are given in Table 12. We can conclude, from the analysis of Table 12, that the proposed algorithm can resist against statistical attacks. We note that the image correlation values of the encrypted image are very close to zero. Table 13 illustrates the distribution of two adjacent pixels.

**Table 12.** Correlation coefficients of two adjacent pixels in the plain/encrypted image.

| | Diagonally | | Horizontally | | Vertically | |
|---|---|---|---|---|---|---|
| | Plain | Ciphered | Plain | Ciphered | Plain | Ciphered |
| **Lena** | 0.8290 | 0.0669 | 0.4454 | −0.0636 | 0.9445 | 0.0465 |
| **Baboon** | 0.8748 | −0.1941 | 0.8834 | −0.1099 | 0.1059 | 0.0487 |
| **Boat** | −0.1561 | 0,2022 | 0.3059 | 0.0149 | 0.1709 | 0.0430 |
| **Clock** | 0.7870 | −0.0919 | 0.887 | −0.1719 | 0.9437 | −0.0314 |
| **Peppers** | 0.6498 | −0.2289 | 0.9315 | 0.2409 | 0.6104 | 0.1633 |
| **Jelly-Beans** | 0.9180 | −0.1200 | 0.9589 | −0.1413 | 0.9726 | 0.0291 |
| **Splash** | 0.8892 | 0.1754 | 0.0108 | 0.0059 | 0.9460 | −0.0499 |

**Table 13.** Distribution of two adjacent pixels in the plain/encrypted images.

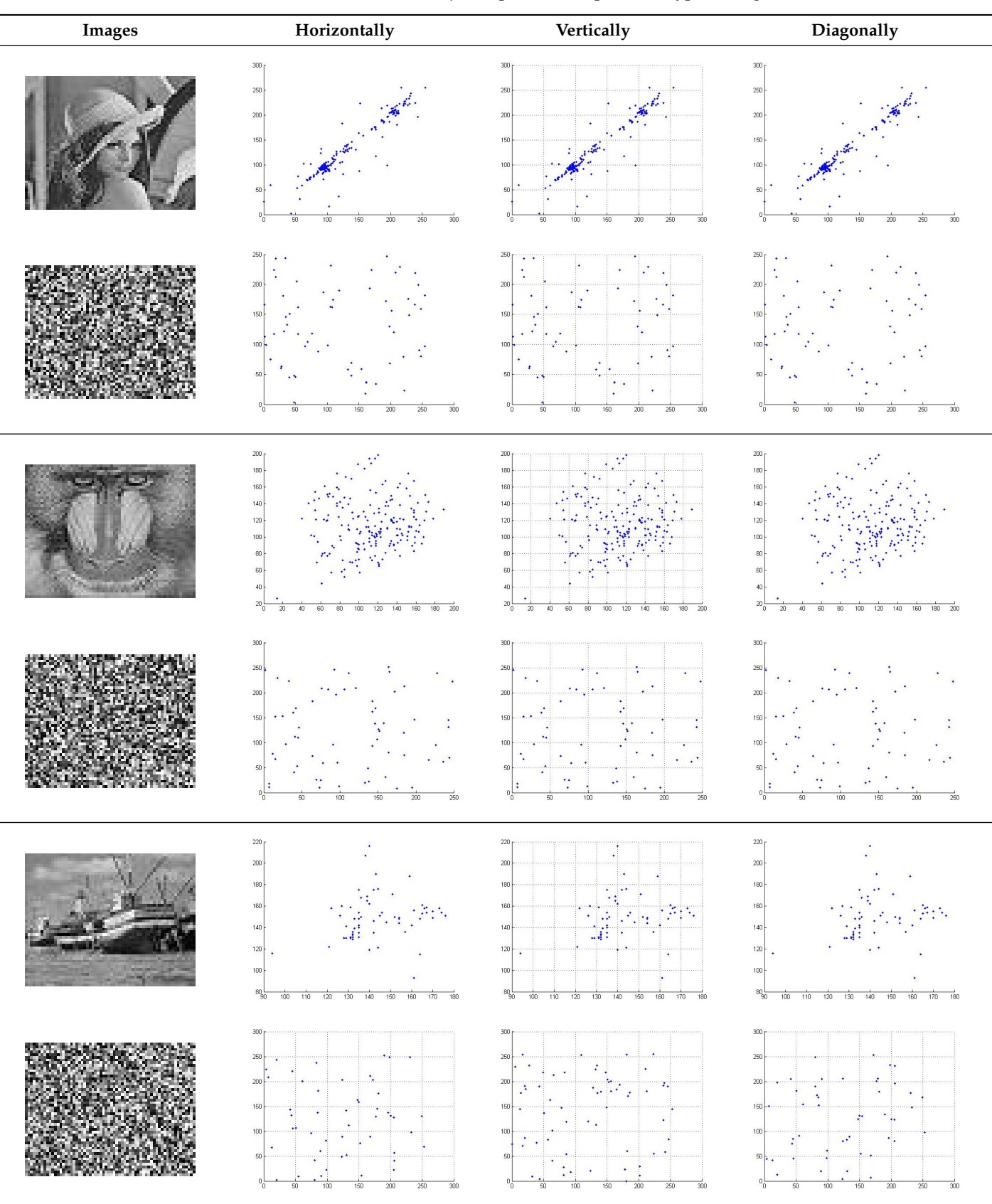

**Table 13.** *Cont*.

| Images | Horizontally | Vertically | Diagonally |
|--------|--------------|------------|------------|

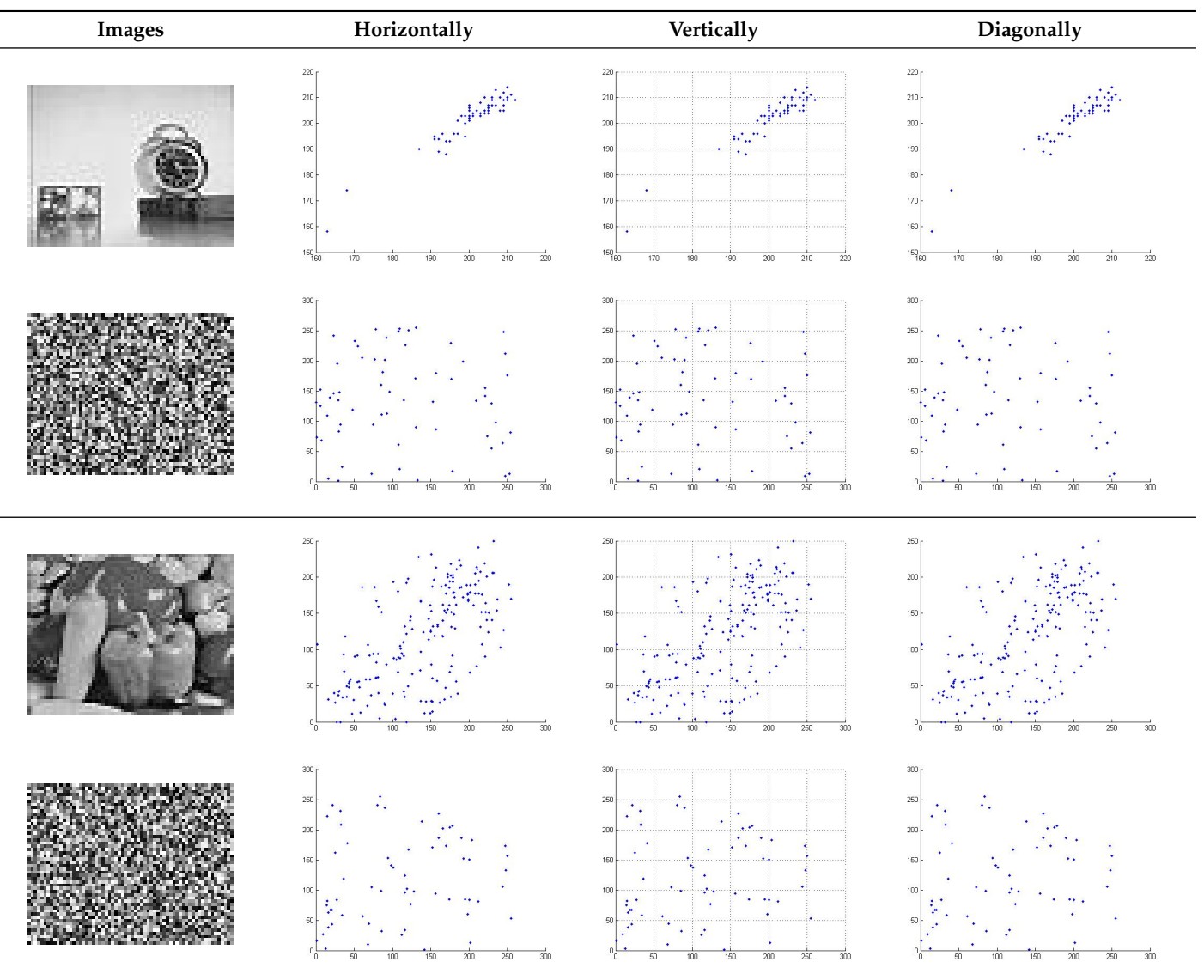

## 8. Conclusions

In this paper, a lightweight encryption algorithm based on the standard AES was proposed. The first step was generating a cryptographically strong S-box based on chaotic Boolean functions. The Hilbert curve scan pattern and the Lorenz system were used in order to realize the permutation and diffusion phases. Cryptographic properties of the chaotic S-box and NIST tests validates its strength. The algorithm was developed to be implemented in highly constrained IoT devices. In order to show the effectiveness of the scheme, it was implemented to encrypt a grayscale image by using an XM1000 sensor. Experimental results showed that the algorithm is light enough to satisfy many criteria like memory consumption, execution time and information entropy.

**Author Contributions:** Supervision and funding acquisition, B.M.A.; writing—original draft and writing—review and editing, R.G.; methodology, T.G.; resources and validation, H.A. and A.A. All authors have read and agreed to the published version of the manuscript.

**Funding:** This research was funded by the Deanship of the Scientific Research of the University of Ha'il, Saudi Arabia (project: RG-20091).

**Conflicts of Interest:** The authors declare no conflict of interest.

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
