# Peer review of "Implementing a Symmetric Lightweight Cryptosystem in Highly Constrained IoT Devices by Using a Chaotic S-Box"

_symmetry, doi:10.3390/sym13010129_

Round 1

Reviewer 1 Report

In this study authors presented a lightweight encrypted algorithm based on the standard AES. Overall, the manuscript is well-written and thoroughly describes the key ideas. However, I would suggest following improvements.

1) The Introduction section is weak, it doesn't clearly explain the motivation of the study. I would highly recommend authors to include paragraph that entails the motivation of study with some real life examples.

2) Related work is added in Introduction. It should be separate section and at the end of Related Work section authors should describe the distinguishing features of this study from existing works.

Reviewer 2 Report

Comments for authors:

In this paper, a lightweight cryptosystem that can be efficiently implemented in highly constrained IOT devices is proposed.
The algorithm is mainly based on Advanced Encryption Standard (AES) and a new chaotic S-box. Since its adoption by the IEEE 802.15.4 protocol, AES in embedded platforms has been increasingly used. The main cryptographic properties of the generated S-box have been validated. The randomness of the generated S-box has been confirmed by the NIST tests.
Experimental results and security analysis demonstrated that the cryptosystem can, on the one hand, reach good encryption results and respects the limitation of the sensor’s resources, on the other hand. So the proposed solution could be reliably applied in image encryption and secure communication between networked smart objects.
Experimental results showed that the algorithm is light enough to satisfy many criteria like memory consumption, execution time and information entropy.

Round 2

Reviewer 1 Report

The authors have improved the manuscript and adequately addressed the comments. I don't have any further comments for this manuscript.

This manuscript is a resubmission of an earlier submission. The following is a list of the peer review reports and author responses from that submission.